# Fast and accurate estimation of the covariance between pairwise maximum likelihood distances

Manuel Gil

Institute of Molecular Life Sciences, University of Zurich, Zurich, Switzerland
Swiss Institute of Bioinformatics, Lausanne, Switzerland

## ABSTRACT

Pairwise evolutionary distances are a model-based summary statistic for a set of molecular sequences. They represent the leaf-to-leaf path lengths of the underlying phylogenetic tree. Estimates of pairwise distances with overlapping paths covary because of shared mutation events. It is desirable to take these covariance structure into account to increase precision in any process that compares or combines distances. This paper introduces a fast estimator for the covariance of two pairwise maximum likelihood distances, estimated under general Markov models. The estimator is based on a conjecture (going back to *Nei & Jin, 1989*) which links the covariance to path lengths. It is proven here under a simple symmetric substitution model. A simulation shows that the estimator outperforms previously published ones in terms of the mean squared error.

## INTRODUCTION

Phylogenetic trees are one of the most important representations of the evolutionary relationship between homologous genomic sequences. Their relatedness can be summarized by a set of pairwise evolutionary distances representing the leaf-to-leaf path lengths of the underlying tree. Such distances are usually estimated by maximum likelihood (ML) assuming a Markovian model of character substitution (*Yang, 2006*).

Besides substitutions, a process of insertions and deletions of sequence fragments plays a major role in the evolution of molecular sequences. As a consequence, homologous characters—i.e., the ones related by substitutions only—have to be identified prior to distance estimation. A consistent hypothesis of character homology is provided by multiple sequence alignments (MSAs). However, they are hard to compute, with non-trivial scoring schemes leading to NP complete problems (*Wang & Jiang, 1994*; *Just, 2001*; *Elias, 2006*). Alternatively, the sequences can be aligned pairwise, for instance, by dynamic programming to obtain optimal pairwise alignments (OPAs) in quadratic time in the length of the input sequences (*Needleman & Wunsch, 1970*).

Pairwise distance methods are generally faster and also simpler than likelihood based approaches that operate directly on sequence data. For that reason, they have often been

Corresponding author
Manuel Gil,
manuel.gil.sci@gmail.com

chosen as an input to large-scale genomic and phylogenetic analyses. Further, distance tree methods are used to produce starting trees for ML tree estimation from MSAs (*Guindon et al., 2010*; *Stamatakis, 2014*; *Vinh & Von Haeseler, 2004*; *Gil et al., 2013*) and guide trees in progressive MSA methods (e.g., *Löytynoja & Goldman, 2008*; *Katoh et al., 2005*).

The speed benefits may affect accuracy due to a potential loss of information involved in the reduction of the sequence data (*Steel, Hendy & Penny, 1988*). However, this idea has recently been challenged in the context of tree estimation (*Roch, 2010*). Roch proposed to take advantage of higher order information using the correlations among the pairwise distances, which result from common mutation events on shared paths of the underlying tree. Indeed, most current practical distance tree methods assume statistical independence and do not account for distance covariance (*Mihaescu & Pachter, 2008*). The BioNJ algorithm (*Gascuel, 1997*), which uses a first-order model of covariance, is a notable exception. Generally, any process that compares or combines distance profits from a higher precision when the covariance structure is taken into account.

Estimators for the covariance between pairwise ML distance estimates have been proposed for certain mechanistic substitution models (*Tajima & Nei, 1984*; *Nei & Jin, 1989*; *Bulmer, 1989*) and for general Markov models by *Susko (2003)*. Susko's estimator requires an MSA and has a linear time complexity in the sequence length. *Dessimoz & Gil (2008)* have derived an adaptation for OPAs with similar complexity. This paper introduces a constant time estimator for general Markov models, applicable to OPAs and MSAs. To this end, a conjecture (from *Nei & Jin, 1989*; *Bulmer, 1989*) for a simple symmetric substitution model is proven and then extended to general models. In a simulation, the estimator is evaluated and compared to previously published ones.

## METHODS

### Preliminaries

Denote $A = \{x_i\}_{i=1}^n$ a pairwise alignment consisting of $n$ homologous i.i.d. character-pairs $x_i$ (e.g., nucleotides, amino acids, or codons, but no insertion–deletions). The likelihood of having the two sequences in $A$ separated by an evolutionary distance $d$ is (*Felsenstein, 1981*)

$$L(A|d) = \prod_{i=1}^n p(x_i, d), \tag{1}$$

where $p(x_i, d)$ is the probability of the character-pair $x_i$ at $d$. The ML estimator of the true distance $d_t$ is

$$\hat{d} = \arg\max_{d \geq 0} L(A|d). \tag{2}$$

While for simple mechanistic substitution models the maximum can be expressed analytically, it is usually found numerically for empirical and complex mechanistic models using the Newton–Raphson method. Let $I_n(d)$ denote the Fisher information for $d$, i.e.,

$$I_n(d) = -nE\left[\frac{\partial^2}{\partial d^2} \log p(X, d)\right], \tag{3}$$

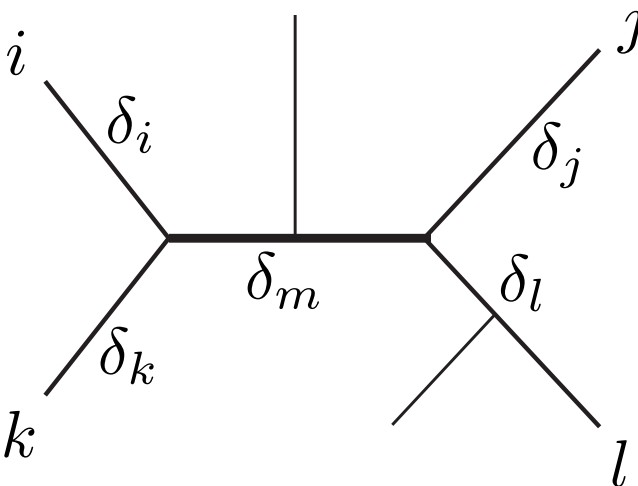

**Figure 1 Unrooted tree relating six sequences.** The labeled sequences $\{i, j, k, l\}$ define a subtree of four sequences, a quartet. The $\delta$'s indicate the branch lengths of the quartet, representing number of character changes. Under Markovian substitution models the ML estimates of the pairwise distances $d_{ij} = \delta_i + \delta_m + \delta_j$ and $d_{kl} = \delta_k + \delta_m + \delta_l$ covary because of the $\delta_m$ common mutation events.

where $X$ is a random variable of which the $x_i$ are realizations. The asymptotic variance of $\hat{d}$ is provided by standard theory (e.g., *Pawitan, 2001*)

$$V(\hat{d}) = I_n(d_t)^{-1}. \tag{4}$$

It can be estimated by evaluating the inverse of the Fisher information at $\hat{d}$ (hereafter *ML variance*)

$$\hat{V}(\hat{d}) = I_n(\hat{d})^{-1}, \tag{5}$$

or, alternatively, from $A$ and $\hat{d}$ by a sample average:

$$\hat{V}_A(\hat{d}) = -\frac{1}{n}\left(\frac{1}{n}\sum_{i=1}^{n}\frac{\partial^2}{\partial d^2}\log p(x_i, d)\bigg|_{d=\hat{d}}\right)^{-1} = -\left(\sum_{i=1}^{n}\frac{\partial^2}{\partial d^2}\log p(x_i, d)\bigg|_{d=\hat{d}}\right)^{-1}. \tag{6}$$

*Dessimoz & Gil (2008)* distinguished three topological relations relevant for covariance estimation between any two pairwise distance estimates. First, the relation *dependence*, where two distances share some common evolution (e.g., $d_{ij}$ and $d_{kl}$ in Fig. 1). Second, the similar relation *triplet*, where two distances additionally share a sequence (e.g., $d_{ij}$ and $d_{jk}$). Third, the case *independence*, where the distances are independent (e.g., $d_{ik}$ and $d_{jl}$). Note that the second case can be conceptually reduced to the first (with $j = l$ and $\delta_j = \delta_l = 0$). Further, we assume that mutation events on different edges of a tree are independent, thus, the distances in the independence-case have zero covariance. Therefore, our derivations will focus on the dependence-case without loss of generality.

General ML theory provides covariance estimates if all unknown parameters are estimated jointly. For instance, in ML tree reconstruction, the variance/covariance matrix

can be estimated from the observed Fisher information matrix. However, the estimation of the pairwise distances considered here is done separately for each distance so that general ML theory can not be applied. *Susko (2003)* has derived an interesting estimator for the covariance of two distances $\hat{d}_{ij}$ and $\hat{d}_{kl}$ estimated from pairwise alignments which are induced by an MSA. It is a sample average based on the following expression (hereafter *Susko-covariance*)

$$\text{cov}(\hat{d}_{ij}, \hat{d}_{kl}) = nV_{ij}V_{kl}E\left[ \frac{\partial}{\partial d} \log p_{ij}(X, d)\bigg|_{d=d_{t,ij}} \cdot \frac{\partial}{\partial d} \log p_{kl}(X, d)\bigg|_{d=d_{t,kl}} \right]. \tag{7}$$

Here, $d_{t,ij}$ and $d_{t,kl}$ are the true distances, the random-variable $X$ stands for quartets of homologous characters (as opposed to pairs in Eq. (3)), and $p_{ij}(X, d)$ denotes the probability for the pair $\{i, j\}$ in $X$ at the distance $d$. The Susko-covariance has two limitations. First, it requires an MSA, i.e., it is not applicable to distances derived from OPAs (for a discussion see *Dessimoz & Gil, 2008*). Second, the O($n$) computation time may become prohibitive in large scale studies. Alternatively, a nonparametric bootstrap can be used (*Efron & Tibshirani, 1993*), but it takes substantially longer computation times and requires an MSA too.

Previously, two estimators have been presented do tackle the limitations. They work with both MSAs and OPAs. The first method is a numerical approximation to the variance of the difference between two distances involving a common sequence. It runs in constant time with respect to the sequence-length and takes the following form (*Dessimoz et al., 2006*)

$$\hat{V}_{NUM}(\hat{d}_{ij} - \hat{d}_{jk}) = \frac{\hat{d}_{ik}^{\phi_1}}{\hat{V}(\hat{d}_{ik})^{\phi_2}} \cdot \frac{\left[\hat{V}(\hat{d}_{ij}) + \hat{V}(\hat{d}_{jk})\right]^{\phi_3}\left[\hat{V}(\hat{d}_{ij})\hat{V}(\hat{d}_{jk})\right]^{\phi_4}}{(\hat{d}_{ij} + \hat{d}_{jk})^{\phi_5}}, \tag{8}$$

where the $\phi_i$ are parameters optimised a priori for a particular substitution model. This leads (through $V(X - Y) = V(X) + V(Y) - 2\text{cov}(X, Y)$) to a fast covariance estimator for the triplet-case (hereafter referred to as *triplet-covariance*)

$$\text{cov}(\hat{d}_{ij}, \hat{d}_{jk}) = \frac{1}{2}\left[\hat{V}(\hat{d}_{ij}) + \hat{V}(\hat{d}_{jk}) - \hat{V}_{NUM}(\hat{d}_{ij} - \hat{d}_{jk})\right]. \tag{9}$$

The second method is based on Susko's theory and shares the linear time complexity (*Dessimoz & Gil, 2008*, hereafter *anchor-covariance*). It was specifically designed to bypass the problem of inconsistent homology inference between OPAs using the concept of *anchors*—a globally consistent subset of aligned character pairs. In this paper, we propose a fast and general approach we term *branch-covariance*. It is motivated by an analytic result obtained under the simple $r$-state symmetric model.

## $r$-state symmetric model

To obtain analytic results we will work with the $r$-state symmetric model, also know as the $N_r$ model (*Neyman, 1971*). It is a generalization of the Jukes–Cantor model (*Jukes &*

*Cantor, 1969*), which has four states ($r = 4$), to $r$ character-states. The $N_r$ model assumes a uniform distribution of states at the root, and equal rates of transitions between any two distinct character states. The probability to observe a mutation after time $t$ is

$$p_m(t) = \beta\left(1 - e^{-\frac{\alpha t}{\beta}}\right), \quad \beta = \frac{r-1}{r}, \tag{10}$$

where $\alpha$ is the total rate of substitution. Thus, if two sequences are separated by $t$, the distance between them will be $d = \alpha t$.

Because of the symmetries in the model, the number of differing sites $I$ in a given pairwise alignment of length $n$ is a sufficient statistics for the pairwise ML distance

$$\hat{d} = -\beta \ln\left(1 - \frac{I}{n\beta}\right). \tag{11}$$

An estimator for the variance of $\hat{d}$ can be obtained by applying Eq. (4) derived from the likelihood function. Alternatively, for models estimating distances from proportions of differing sites, the variance can be approximated by the Delta technique. This has been done by *Kimura & Ohta (1972)* for $r = 4$ and generalized by *Tajima & Nei (1984)* to

$$\hat{V}(\hat{d}) = \beta\left[(1 - \beta)e^{\frac{2d}{\beta}} + (2\beta - 1)e^{\frac{d}{\beta}} - \beta\right]/n. \tag{12}$$

We are going to use the Delta technique to derive an estimator for the covariance of two $N_r$ ML distances $d_{ij} = \delta_i + \delta_m + \delta_j$ and $d_{kl} = \delta_k + \delta_m + \delta_l$ in the dependence-case (Fig. 1). *Nei & Jin (1989)* used an informal argument to propose the following expression with $\beta = 3/4$:

$$\text{cov}(\hat{d}_{ij}, \hat{d}_{kl}) = \beta\left[(1 - \beta)e^{\frac{2\delta_m}{\beta}} + (2\beta - 1)e^{\frac{\delta_m}{\beta}} - \beta\right]/n. \tag{13}$$

The equation originates from the assumption, that the covariance of two distance estimates with an underlying shared path length $\delta_m$, is formally equivalent to the variance (Eq. (12)) of an ML estimate of a pairwise distance $\delta_m$. Indeed, *Bulmer (1989)* presented a proof for $\beta = 3/4$ for the triplet-case and conjectured that Eq. (13) with $j = l$ and $\delta_j = \delta_l = 0$ is true for any $\beta$. To the best of our knowledge Eq. (13) has not been proven yet in its most general form, i.e., for the dependence-case and any $\beta$.

We will first compute

$$\text{cov}(I_{ij}, I_{kl}) = n\text{cov}(S_{ij}, S_{kl}) = nE[S_{ij}S_{kl}] - nE[S_{ij}]E[S_{kl}], \tag{14}$$

where $S_{ij}$ is a random variable indicating whether sequences $i$ and $j$ are identical ($S_{ij} = 0$) or different ($S_{ij} = 1$) at a particular site. Subsequently, we will apply the Delta method to obtain Eq. (13) from $\text{cov}(I_{ij}, I_{kl})$. We start by noting that

$$E[S_{ij}] = \text{Pr}(S_{ij} = 1) = p_m(d_{ij}). \tag{15}$$

Therefore, the problem reduces to computing

$$E[S_{ij}S_{kl}] = \Pr(S_{ij} = 1 \wedge S_{kl} = 1). \tag{16}$$

In the following we will represent the quartet from Fig. 1 by the symbol $\succ\!\!\!-\!\!\!\prec$, where the terminal node of the upper left branch corresponds to $i$. Furthermore, we are going to mark a branch with $\circ$ if a particular site in the evolving sequence is in a different state at the endpoints of the branch. As an example, we look at the pattern $\succ\!\!\!-\!\!\!\prec$. Here, the site in question changed its state on the branches leading to $\{j, k, l\}$ but did not change its state on the branch leading to $i$ and on the middle branch. A particular labeling of the nodes with characters from the alphabet of $N_r$ for the given pattern has probability

$$\Pr(\succ\!\!\!-\!\!\!\prec) = \frac{1}{r} \cdot (1 - p_m(\delta_i)) \cdot \frac{p_m(\delta_j)}{(r-1)} \cdot \frac{p_m(\delta_k)}{(r-1)} \cdot \frac{p_m(\delta_l)}{(r-1)} \cdot (1 - p_m(\delta_m)). \tag{17}$$

For this pattern, there are $r(r-1)^3$ possible labellings, of which only $r(r-1)^2(r-2)$ satisfy $S_{ij} = 1 \wedge S_{kl} = 1$. Symmetrical mutation patterns, like for example $\succ\!\!\!-\!\!\!\prec$ have the same number of labelings leading to $S_{ij} = 1 \wedge S_{kl} = 1$ but different mutation probabilities. We consider now all the patterns (grouped by symmetry) and corresponding labelings for the desired condition:

$$\begin{aligned}
&\Pr(S_{ij} = 1 \wedge S_{kl} = 1) \\
&= \Pr(\succ\!\!\!-\!\!\!\prec) \cdot r(r-1) \\
&\quad + [\Pr(\succ\!\!\!-\!\!\!\prec) + \Pr(\succ\!\!\!-\!\!\!\prec) + \Pr(\succ\!\!\!-\!\!\!\prec) + \Pr(\succ\!\!\!-\!\!\!\prec)] \cdot r(r-1)^2 \\
&\quad + [\Pr(\succ\!\!\!-\!\!\!\prec) + \Pr(\succ\!\!\!-\!\!\!\prec) + \Pr(\succ\!\!\!-\!\!\!\prec) + \Pr(\succ\!\!\!-\!\!\!\prec)] \cdot r(r-1)(r-2) \\
&\quad + [\Pr(\succ\!\!\!-\!\!\!\prec) + \Pr(\succ\!\!\!-\!\!\!\prec) + \Pr(\succ\!\!\!-\!\!\!\prec) + \Pr(\succ\!\!\!-\!\!\!\prec)] \cdot r(r-1)^2(r-2) \\
&\quad + [\Pr(\succ\!\!\!-\!\!\!\prec) + \Pr(\succ\!\!\!-\!\!\!\prec) + \Pr(\succ\!\!\!-\!\!\!\prec) + \Pr(\succ\!\!\!-\!\!\!\prec)] \cdot r(r-1)(r-2)^2 \\
&\quad + \Pr(\succ\!\!\!-\!\!\!\prec) \cdot r(r-1)^2(r-2)^2 \\
&\quad + [\Pr(\succ\!\!\!-\!\!\!\prec) + \Pr(\succ\!\!\!-\!\!\!\prec)] \cdot r(r-1)\left[(r-1) + (r-2)^2\right] \\
&\quad + [\Pr(\succ\!\!\!-\!\!\!\prec) + \Pr(\succ\!\!\!-\!\!\!\prec) + \Pr(\succ\!\!\!-\!\!\!\prec) + \Pr(\succ\!\!\!-\!\!\!\prec)] \\
&\qquad \cdot r(r-1)\left[(r-1) + (r-2)^2\right](r-2) \\
&\quad + \Pr(\succ\!\!\!-\!\!\!\prec) \cdot r(r-1)\left[(r-1) + (r-2)^2\right]^2. \tag{18}
\end{aligned}$$

Using Maple (script in Supplemental Materials) we find that the expression simplifies to

$$E[S_{ij}S_{kl}] = \beta \left(1 - \beta e^{-\frac{d_{ij}}{\beta}} - \beta e^{-\frac{d_{kl}}{\beta}} + (1-\beta)e^{-\frac{d_{ij}+d_{kl}-2\delta m}{\beta}} + (2\beta - 1)e^{-\frac{d_{ij}+d_{kl}-\delta m}{\beta}}\right). \tag{19}$$

Plugging Eqs. (19) and (15) in (14) we obtain

$$\mathrm{cov}(I_{ij}, I_{kl}) = n\beta \left((1-\beta)e^{-\frac{d_{ij}+d_{kl}-2\delta m}{\beta}} + (2\beta - 1)e^{-\frac{d_{ij}+d_{kl}-\delta m}{\beta}} - \beta e^{-\frac{d_{ij}+d_{kl}}{\beta}}\right). \tag{20}$$

We turn to the Delta method. The function $\hat{d}_{ij}(I_{ij})$ can be approximated by a first-order Taylor series around $E[I_{ij}] = np_m(d_{ij})$

$$\hat{d}^{\star}_{ij}(I_{ij}) = \hat{d}_{ij}(E[I_{ij}]) + \hat{d}'_{ij}(E[I_{ij}])(I_{ij} - E[I_{ij}]), \tag{21}$$

where

$$\hat{d}'_{ij}(E[I_{ij}]) = \left.\frac{\partial \hat{d}_{ij}(I_{ij})}{\partial I_{ij}}\right|_{I_{ij}=E[I_{ij}]} = \left.\left(n - \frac{I_{ij}}{\beta}\right)^{-1}\right|_{I_{ij}=E[I_{ij}]} = \frac{e^{\frac{d_{ij}}{\beta}}}{n}. \tag{22}$$

The covariance of $\hat{d}_{ij}$ and $\hat{d}_{kl}$ is asymptotically equal to the covariance of $\hat{d}^{\star}_{ij}$ and $\hat{d}^{\star}_{kl}$

$$\text{cov}\left(\hat{d}_{ij}(I_{ij}), \hat{d}_{kl}(I_{kl})\right) \sim \text{cov}\left(\hat{d}^{\star}_{ij}(I_{ij}), \hat{d}^{\star}_{kl}(I_{kl}))\right) \tag{23}$$

$$= \hat{d}'_{ij}(E[I_{ij}])\hat{d}'_{kl}(E[I_{kl}])\text{cov}(I_{ij}, I_{kl}) \tag{24}$$

$$= \beta\left[(1-\beta)e^{\frac{2\delta_m}{\beta}} + (2\beta-1)e^{\frac{\delta_m}{\beta}} - \beta\right]/n. \quad \square \tag{25}$$

We provide a Maple program implementing all steps of the proof in Supplemental Materials.

## Covariance under general Markov models

We have derived under $N_r$ that the covariance of two distance estimates with an underlying shared path length $\delta_m$ is asymptotically equal to the variance of an ML estimate of a true pairwise distance $\delta_m$, i.e.,

$$\text{cov}(\hat{d}_{ij}, \hat{d}_{kl}) \sim V(\hat{d}|d_t = \delta_m). \tag{26}$$

We conjecture now that the relationship holds for general substitution models and apply it to derive a covariance estimator. To this end, we first discuss how the covariance can be written in terms of a rate matrix $Q$, an equilibrium frequency vector $\pi$ (which together fully specify a substitution model), and $\delta_m$. In the next section, we will show how $\delta_m$ can be estimated from the input distances by the method of weighted least squares (WLS).

According to the conjecture and Eq. (4) we express the desired covariance by

$$\text{cov}(\hat{d}_{ij}, \hat{d}_{kl}) = -\frac{1}{n}E\left[\left.\frac{\partial^2}{\partial d^2}\log p(X,d)\right|_{d=\delta_m}\right]^{-1}. \tag{27}$$

The expected value expression can be written as

$$\sum_{\forall(u,v)}\left[p(x_{uv},d)\frac{\partial^2}{\partial d^2}\log p(x_{uv},d)\right]_{d=\delta_m} = \sum_{\forall(u,v)}\left[\frac{\partial^2}{\partial d^2}p(x_{uv},d) - \frac{\left(\frac{\partial}{\partial d}p(x_{uv},d)\right)^2}{p(x_{uv},d)}\right]_{d=\delta_m}, \tag{28}$$

where the summation goes over all possible character pairs. In terms of a rate matrix $Q$ and an equilibrium frequency vector $\pi$ the probability of a character pair ($k = 0$) and the

derivatives ($k > 0$) of the probability with respect to the distance are

$$\frac{\partial^{(k)}}{\partial d^{(k)}} p(x_{uv}, d) = \pi_u \left[ Q^k e^{Qd} \right]_{uv}. \tag{29}$$

Plugging Eqs. (28) and (29) in (27) we obtain

$$\text{cov}(\hat{d}_{ij}, \hat{d}_{kl}) = -\frac{1}{n} \left[ \sum_{\forall (u,v)} \pi_u \left( \left[ Q^2 e^{Q\delta_m} \right]_{uv} - \frac{\left[ Q e^{Q\delta_m} \right]^2_{uv}}{\left[ e^{Q\delta_m} \right]_{uv}} \right) \right]^{-1}. \tag{30}$$

A plot of this expression as a function of $\delta_m$ can be found in Fig. S1. A covariance estimator is obtained by substituting $\delta_m$ in Eq. (30) by its WLS estimate, which we derive in the next section. To save computation time, we can discretise the distance space in the relevant range to some desired level of accuracy, precompute the expected value expressions, and store them in a hash table.

## Topological relation and path length

Equation (30) expresses the covariance in the dependence case as a function of the shared path length $\delta_m$. To obtain a covariance estimator, we determine first whether the two distances in question are dependent and, in case they are, estimate $\delta_m$. We will do that by WLS using the six pairwise distance estimates $\{\hat{d}_{uv}\}$ between the four sequences $i, j, k, l$ and their variances $\{v_{uv}\}$. The sequences can be related by three topological configurations:

$$T_1 : \big((i,k),(j,l)\big), \qquad T_2 : \big((i,l),(j,k)\big), \qquad T_3 : \big((i,j),(k,l)\big),$$

where $T_1$, $T_2$ map to the dependence case and $T_3$ corresponds to the independence case. An argument set out in Supplemental Materials shows that the weighted sum of squares ($S$) for each of the topologies can be expressed in a simple form which is fast to compute:

$$S(T_1) = \frac{\hat{d}_{ij} + \hat{d}_{kl} - \hat{d}_{il} - \hat{d}_{jk}}{v_{ij} + v_{kl} + v_{il} + v_{jk}}, \qquad S(T_2) = \frac{\hat{d}_{ij} + \hat{d}_{kl} - \hat{d}_{ik} - \hat{d}_{jl}}{v_{ij} + v_{kl} + v_{ik} + v_{jl}},$$
$$S(T_3) = \frac{\hat{d}_{ik} + \hat{d}_{jl} - \hat{d}_{il} - \hat{d}_{jk}}{v_{ik} + v_{jl} + v_{il} + v_{jk}}. \tag{31}$$

The best fitting topology is determined by $\text{argmin}_{T_i}\{S(T_i)\}$. If this results in $T_3$ the desired covariance is zero, otherwise we need to estimate $\delta_m$. The WLS estimates are (see Supplemental Materials):

$$2\hat{\delta}_m(T_1) = \frac{(\hat{d}_{ij} + \hat{d}_{kl})(v_{il} + v_{jk}) + (\hat{d}_{il} + \hat{d}_{jk})(v_{ij} + v_{kl})}{v_{ij} + v_{kl} + v_{il} + v_{jk}} - (\hat{d}_{ik} + \hat{d}_{jl}), \tag{32}$$

$$2\hat{\delta}_m(T_2) = \frac{(\hat{d}_{ij} + \hat{d}_{kl})(v_{ik} + v_{jl}) + (\hat{d}_{ik} + \hat{d}_{jl})(v_{ij} + v_{kl})}{v_{ij} + v_{kl} + v_{ik} + v_{jl}} - (\hat{d}_{il} + \hat{d}_{jk}). \tag{33}$$

Since these are the estimators for unconstrained WLS they can result in negative values, in which case we estimate the covariance to be zero.

## Simulation settings

The performance of the various covariance estimators was evaluated with the same simulation approach as in a previous study (*Dessimoz & Gil, 2008*). Specifically, 100 random quartets were sampled from a tree of life on 352 species. The tree was inferred by the *LeastSquaresTree* function (*Gil & Gonnet, 2009*) included in the Darwin package (*Gonnet et al., 2000*) using pairwise distance and variance data from the OMA project (*Dessimoz et al., 2005*). A uniformly distributed $U(0.5, 2)$ expansion/contraction factor was applied on each quartet to also explore extremer regions of the branch-length space, while preserving the relative branch-length structure of the original tree.

For each dilated model quartet 10,000 times three random amino-acid sequences of length $m = \{200, 500, 800\}$ were generated and mutated along the quartet assuming the GCB substitution model (*Gonnet, Cohen & Benner, 1992*). The entire simulation procedure was run twice, once without any insertion–deletions to produce ungapped alignments to test the methods under the true models (i.e., without the effect of alignment errors), and once introducing gaps of Zipfian distributed length (*Benner, Cohen & Gonnet, 1993*). The gapped sequences were aligned by global pairwise dynamic programming with the *Align* function from Darwin to obtain OPAs. ML pairwise distances were estimated with the *EstimatePam* function. The sample variance and covariance over the 10,000 samples (hereafter *Monte Carlo-variance* and *Monte Carlo-covariance*) served as a reference values, as they are unbiased estimators of the true values.

To test the asymptotic conjecture (Eq. (26)) the ungapped simulation was repeated with very long sequences (10,000 amino-acids). Then, the length $\delta_m$ of the middle branch of each model quartet was extracted. Subsequently, 10,000 random sequences were generated, and mutated each with a distance $\delta_m$ according to the GCB model. Finally, the ML distance and ML variance between each resulting pair of sequences, and the Monte Carlo-variance were computed.

## RESULTS AND DISCUSSION

### Evaluation of basic components of branch-covariance

We have tested the validity of the conjecture in Eq. (26), which was derived under the $N_r$ model, and the accuracy of the ML variance (Eq. (5)) in a simulation under the GCB model with long sequences (10,000 amino-acids).

A plot of the Monte Carlo-variance of the ML estimate of a pairwise distance $\delta_m$ versus the Monte Carlo-covariance between the ML estimates of two pairwise distances in the dependence case with a shared path of length $\delta_m$ corroborates the conjecture and suggests that the result is valid in general (Fig 2A). The branch-covariance relies on the ML variance to approximate the true variance. A comparison with the Monte Carlo-variance shows that it underestimates the variance for large samples and with correct alignments (Fig 2B). Therefore, we expected the branch-covariance to inherit the negative bias.

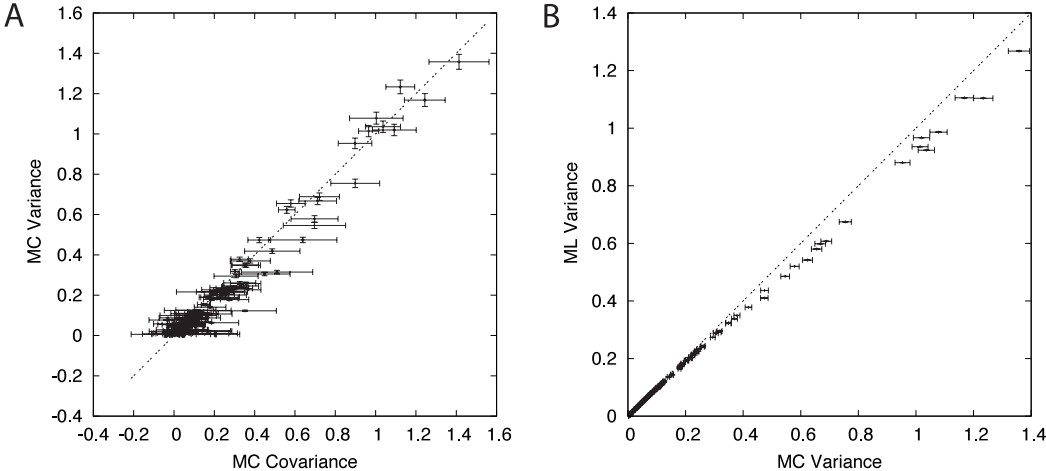

**Figure 2 Components of branch-covariance for sequence-length 10,000 amino-acids.** Error-bars indicate 95% confidence intervals. (A) Monte Carlo covariance versus Monte Carlo variance shows that the relationship derived under the $N_r$ model also holds for the empirical substitution model tested here. (B) Monte Carlo variance versus mean of the ML variance shows that the ML variance is negatively biased.

## Evaluation of estimators

To evaluate and compare the various estimators under ideal conditions we have carried out two types of simulations: one without introducing indel events (ungapped simulation), and one with indel events (gapped simulation).

The gapped simulation did not require explicit alignment; the MSA is trivial without gaps. This situation corresponds to the model assumed by the Susko-, branch-, and triplet-covariance. Note that in practice with real data, there are usually gaps in an MSA estimate. In this case, the distance and covariance estimators are applied assuming the MSA to be correct. Gaps are either excluded from the analysis, or treated as unknown characters.

The ungapped simulation required alignment which was carried out by OPA. Here, the Susko-covariance can not be applied; it requires an MSA. The setting corresponds to the model assumed by the anchor-covariance, which is an adaptation of the Susko-covariance, specifically designed to bypass the problem of inconsistent homology inference between OPAs. The triplet- and branch-covariance are applicable to OPAs, as these estimators operate directly on the distances and not on the alignments (but the assumption of a gap-free and correct underlying alignment is violated).

### *Ungapped simulation*

This section presents the performance of the branch-covariance under the true model and compares it with the Susko- and triplet-covariance (Fig. 3). To this end, the three topological cases—dependence, independence, and triplet—are treated separately.

In the dependence case the Susko-covariance is unbiased. The branch-covariance lies in most cases within the 95% confidence interval of the Monte Carlo covariance; when it lies outside then it underestimates. The negative bias confirms the prediction from the previous section. In the independence case, where the true covariance is zero, both estimators have a positive bias of comparable magnitude, though the branch-covariance

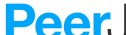

**Figure 3 Ungapped simulation.** Comparison of Suko-covariance (red), branch-covariance (green) and triple-covariance (blue) with their Monte Carlo counterpart for sequence lengths of {200, 500, 800} amino-acids. Error-bars indicate 95% confidence intervals.

**Table 1 Ungapped simulation.** Average MSE, median MSE (italic), and average of ratios between MSE of branch-covariance and MSE of Susko-covariance (bold). A ratio of 0.16, for instance, indicates that the MSE of the branch-covariance was—on average—16% of the MSE of the Susko-covariance. Dependence (D), triplet (T) and independence (I) case for sequence lengths 200, 500 and 800 amino-acids.

| | | 200 | | | 500 | | | 800 | | |
|---|---|---|---|---|---|---|---|---|---|---|
| | | **D** | **T** | **I** | **D** | **T** | **I** | **D** | **T** | **I** |
| **Branch** | Avg | 39.89 | 255.86 | 2.37 | 3.13 | 16.95 | 0.12 | 0.71 | 4.73 | 0.03 |
| | *Median* | *17.94* | *72.63* | *0.45* | *1.29* | *5.37* | *0.01* | *0.33* | *1.35* | *0.00* |
| **Susko** | Avg | 215.96 | 420.72 | 125.59 | 12.74 | 22.47 | 8.06 | 2.76 | 5.19 | 1.87 |
| | *Median* | *112.93* | *121.81* | *61.47* | *6.67* | *7.65* | *3.54* | *1.56* | *2.00* | *0.79* |
| | **Avg ratio** | **0.16** | **0.80** | **0.07** | **0.17** | **0.70** | **0.01** | **0.27** | **0.78** | **0.02** |

appears to have a lower bias with increasing sequence length. For the anchor-covariance the negative bias is no surprise; it returns by construction non-negative values. For the triplets, as in the dependence case, Susko's estimator appears to be unbiased and the branch-covariance shows a minor negative bias. The triplet-covariance has a positive bias of comparable magnitude to the one of the branch-covariance. Although the lack of bias is an attractive feature of an estimator, it does not guarantee a low total error, the sum of systematic and random error. Therefore, it is instructive to look at the mean square error (MSE). Indeed, the branch-covariance has a lower standard deviation and MSE than the Susko-covariance under all three topological relations (Fig. S2, Table 1).

### Gapped simulation

The branch-, anchor-, and triplet-covariance have been tested on distances derived from OPAs.

The branch-covariance has a lower bias than the anchor-covariance for all three topological relations (Fig. 4). In the dependence case the differences are minor, though the branch-covariance has a consistently higher correlation with the Monte Carlo covariance. A big difference is visible for the two other topological relations (independence and triplet), where the anchor-covariance's bias is up to twice the branch-covariance's. The superiority of the branch-covariance is also reflected by the MSEs (Table 2). The triplet-covariance has a greater bias than the branch-covariance for sequences of length 200; for length 500 the two estimators have quantitatively a similar bias, but in opposite directions; and for length 800 the triplet-covariance has clearly a smaller systematic error.

Note that the anchor- and triplet-covariance have been evaluated under the same simulation conditions in a previous work (*Dessimoz & Gil, 2008*). They have been reproduced to evaluate the branch-covariance. The results on the triplet- and anchor-covariance reported here are in agreement with our previous results.

## CONCLUSION

A fast and general method to estimate the covariance of pairwise ML distances estimates has been presented. The estimator is based on a conjecture (going back to *Nei & Jin, 1989*) which links the covariance to path lengths on the underlying phylogenetic tree.

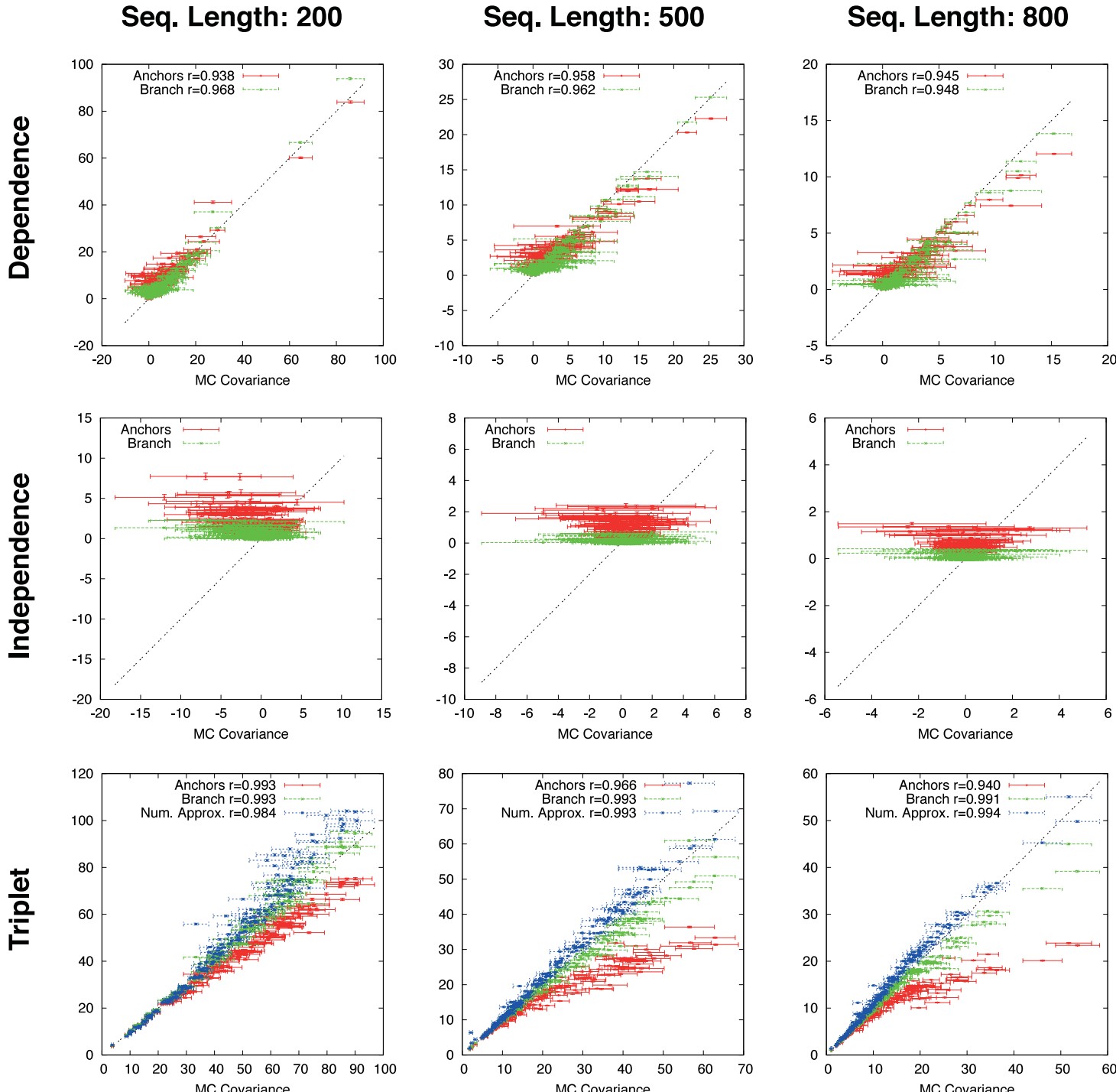

**Figure 4 Gapped simulation.** Comparison of anchor-covariance (red), branch-covariance (green) and triplet-covariance (blue) with their Monte Carlo counterpart for sequence lengths of {200, 500, 800} amino-acids. Error-bars indicate 95% confidence intervals.

**Table 2 Gapped simulation.** Average MSE, median MSE (italic), and average of ratios between MSE of branch-covariance and MSE of anchor-covariance (bold). A ratio of 0.22, for instance, indicates that the MSE of the branch-covariance was—on average—22% of the MSE of the anchor-covariance. Dependence (D), triplet (T) and independence (I) case for sequence lengths 200, 500 and 800 amino-acids.

| | | 200 | | | 500 | | | 800 | | |
|---|---|---|---|---|---|---|---|---|---|---|
| | | D | T | I | D | T | I | D | T | I |
| **Branch** | Avg | 36.19 | 170.04 | 3.63 | 3.99 | 27.23 | 0.21 | 1.38 | 9.68 | 0.06 |
| | *Median* | *19.13* | *110.60* | *1.23* | *1.95* | *6.26* | *0.03* | *0.48* | *1.88* | *0.00* |
| **Anchor** | Avg | 159.07 | 242.56 | 95.79 | 11.60 | 95.62 | 7.68 | 3.19 | 42.87 | 2.09 |
| | *Median* | *102.13* | *170.27* | *59.02* | *6.66* | *12.30* | *3.51* | *1.67* | *4.55* | *0.92* |
| | **Avg ratio** | **0.22** | **0.67** | **0.06** | **0.22** | **0.36** | **0.06** | **0.20** | **0.34** | **0.04** |

It has been proven here under a simple symmetric substitution model and formulated for general models. The estimator is applicable to distances estimated from parametric as well as empirical substitution models, and works with both MSAs and OPAs. A simulation has shown that the estimator has a lower total error than previously published estimators operating directly on the sequence data. Moreover, in contrast to these linear time complexity methods, the estimator runs in constant time in the sequence-length, provided that the pairwise distance information has been precomputed (as in the initial all-against-all pairwise comparison, customary to most pairwise distance approaches).

The evaluation under the correct model conducted here is an important baseline. However, ideal conditions are never met when working with real data, so that as future work the various estimators should be compared in situations where the model-assumptions are violated. Under such conditions, it is conceivable that estimators operating directly on the sequence data outperform the method presented here. The rationale being that they extract information from the data at hand—for instance, the actual frequencies of aligned character pairs, as opposed to the ones assumed in the substitution model—and by being more adaptive, they could be more robust to model violations.

### Funding

This work was supported by the SNF grant PBEZP2_140129 to MG and the ERC starting grant UMICIS (242870) to Christian von Mering. The funders had no role in study design, data collection and analysis, decision to publish, or preparation of the manuscript.

### Grant Disclosures

The following grant information was disclosed by the author:
SNF: PBEZP2_140129.
ERC Starting: UMICIS (242870).

### Competing Interests

The author declares there are no competing interests.

## Author Contributions

- Manuel Gil conceived and designed the experiments, performed the experiments, analyzed the data, contributed reagents/materials/analysis tools, wrote the paper, prepared figures and tables, reviewed drafts of the paper.

## Supplemental Information

Supplemental Materials for this article can be found online at http://dx.doi.org/10.7717/peerj.583#supplemental-information. It consists of a 4-page PDF file and a ZIP file. The PDF file contains (1) the MAPLE code supporting the proof in Section "r-state symmetric model"; (2) the MAPLE code deriving equations (31)–(33); (3) Figures S1 and S2. The ZIP file contains (1) A file Readme.txt describing the simulation environment and the format of the results; (2) the Darwin code for the simulation; (3) the raw results of the simulation.

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
