# Peer review of "Fast and accurate estimation of the covariance between pairwise maximum likelihood distances"

_PeerJ, doi:10.7717/peerj.583_

## Round 0.1 · original submission · Minor Revisions

I suggest the author to address reviewers comments that improve the quality of the manuscript or detailing reasons why certain comments are not addressed. Please provided a point-by-point reply to the reviewers comments to accompany your revised submission and clearly indicate the changes in the manuscript.

Reviewer 1 ·

Basic reporting

The manuscript by Manuel Gil introduce a novel covariance estimator of pairwise
distances for biological sequences. The mathematical derivations are correct
and adequately described and the limitations of the estimator are honestly
discussed. The simplicity of its interpretation as well as the computational
efficiency to compute the estimator makes it a valuable contribution for the
community.
The manuscript is written in a concise manner.

Experimental design

The experiments to validate the novel estimator are adequate and have been conducted rigorously.

Validity of the findings

The statistical analysis is sound. All conclusions are well supported with results.

Additional comments

I have only a few minor
comments and suggest hereby to accept the manuscript once they have been
addressed.

- page 1: You should provide a reference for the clam that OPAs can be computed
more efficiently than MSAs.

- Eq 7: This formula is unclear to me. How are hat(d_i), hat(d_j), d_{t,i} and
d_{t,j} defined? Shouldn't that be hat(d_{i,k}), hat(d_{k,l}), d_{i,j} and
d_{k,l}? Otherwise, you might want to elaborate a bit on this formula.

- typo in last sentence: '... and *an* requires an MSA too.'

- It might be worth to also include the formula for the triplet covariance
somewhere here for the sake of completeness.

- page 7 (Evaluation of basic components... and Figure 2A): Are you measuring
the MC var and MC cov from the delta_m part only? Otherwise, it is for me not
obvious why the two measures should be equal.

- Figure 2-4: These figures are have the same x and y scales. It might be
helpful to create them as square plots. Furthermore, from the caption/figure
it is not clear what is the difference between Figure 3 and 4. I suggest
adding a title or extending the caption to indicate 'based on MSA/OPAs'.

- Are the results in Figure 3/4 based on the gapped or ungapped simulations? I
think it is not written in the text.

- Do you have an explanation for the result that the Triplet configuration has
a six fold higher MSE for your estimator than in the dependence case. For the
Susko estimator this difference is much less pronounced.

- Table 1/2: Some visual support (e.g. colors or vertical lines) to separate
the 3 sequence length categories might be helpful.

·

Basic reporting

I am a little bit confused about the use of the distance symbol d. On page 2 the author uses d_{ij} to indicate the distance between leaves i and j. But further down we find the symbol \hat{d}_i which now indicates a distance measure. I suggest to (a) use an index different from i for the measure and maybe use it as an superscript rather than a subscript to avoid confusion.

As this is a single author paper, the author could use "I" instead of "we" when he describes his steps and approaches.

Experimental design

The design starts with the equations. Is it possible to provide plots of values from Equation (28) to indicate the dependence of covariance to the length of the interior edge? In the simple model, Q is fixed as 1/r for all off-diagonal elements if I am not mistaken, so this shouldn't be a problem.

This should immediately give an idea how the interior edge will influence the range of the MC covariance simulated for Table 1. Because here my next problem arises. The differences in MSE between the Susko method and the branch-covariance method appear MASSIVE, which makes me wonder if there is something wrong.

It appears that large MC covariances are not observed very often. This can have two reasons: (1) Large MC covariances are rare, or (2) The simulation design was biased against selecting examples with large MC covariances. If the author can propose that their simulation design rarely found large MC covariances then I am satisfied with (1), otherwise I suggest the author tries to adapt his design to obtain a better sampling of the covariances. I think that it is likely that (1) can be shown by simply plotting Equation (28). If the covariances go asymptotically (with increasing \delta_m) against a fixed value this might explain this pattern. Also, would it be possible to use the MC correlation instead of the covariance? Since the correlation provides an upper bound to potential values the expected asymptotical behaviour mentioned above could be described better.

The asymptotic equality of \hat{d}_{ij} and \hat{d}_{ij}^\ast: Is this asymptotic in the sequence length, n? And if so, can we get an idea on how fast this asymptote works? If the difference is negligible for n>200 we are good, but if we need n>200,000 for a reasonable equality this approximation might have a problem with applicability. I think this can be easily done by inserting a couple of values for \beta and \delta_m and study the differences as n increases. The main factor for this is time, sadly.

Validity of the findings

I like the Maple-supported proof of the equalities. I couldn't find any mistakes in the methodology, good work.

Looking at the plot in S3 it seems that both MSE's are strongly right skewed making the mean a bad measure of centre. Using the median instead might give a better way of comparing the results. Also, considering the increasing distance between Susko MSE and branch MSE with increasing MC covariance (in both plots), it might be more informative to report on the ratio of errors between the two methods. E.g., if the branch MSE is consistently half the Susko MSE this is much more informative than the mean.

Additional comments

I like the overall work in the paper and find it worth publishing, the work proposed for the future can help speed up large-scale analyses as the genetic data we are receiving from the new technology are making MSAs harder to generate while OPAs are still achievable.

I finish the review with a list of typos I discovered:
Page 3, end of third like after Eq (7): t missing at "not"
Page 6, fourth line from below: "quartet", not "quarted"
Page 7, second line below Evaluation of estimators: "three topological cases" instead of "tree topological cases"?
Page 8, second line below Application to optimal pairwise alignments: "an adaptation of the Susko-covariance".

---

## Round 0.2 · accepted · Accept

Thank you for thoroughly addressing all reviewers comments.